# Kynureninase Upregulation Is a Prominent Feature of NFR2-Activated Cancers and Is Associated with Tumor Immunosuppression and Poor Prognosis

**DOI:** 10.3390/cancers15030834

**Published:** 2023-01-29

**Authors:** Ricardo A. León-Letelier, Ali H. Abdel Sater, Yihui Chen, Soyoung Park, Ranran Wu, Ehsan Irajizad, Jennifer B. Dennison, Hiroyuki Katayama, Jody V. Vykoukal, Samir Hanash, Edwin J. Ostrin, Johannes F. Fahrmann

**Affiliations:** 1Department of Clinical Cancer Prevention, The University of Texas MD Anderson Cancer Center, Houston, TX 77030, USA; 2Department of Biostatistics, The University of Texas MD Anderson Cancer Center, Houston, TX 77030, USA; 3Department of General Internal Medicine, The University of Texas MD Anderson Cancer Center, Houston, TX 77030, USA

**Keywords:** NRF2, KYNU, immunosuppression, prognosis, multi-cancer

## Abstract

**Simple Summary:**

We previously reported a novel mechanism of NRF2 tumoral immune suppression through selective upregulation of the tryptophan-metabolizing enzyme kynureninase (KYNU) in lung adenocarcinoma. In the current study, we report that elevated tumoral KYNU gene expression is broadly associated with NRF2-activated cancers, including pancreatic adenocarcinoma (PDAC). Mechanistic studies confirmed that NRF2 regulates the expression of enzymatically functional KYNU in PDAC cells. We further demonstrated that elevated tumoral KYNU mRNA expression is associated with an immunosuppressive tumor microenvironment across several cancer types and is prognostic for poor overall survival in thymoma, acute myeloid leukemia, low-grade glioma, kidney renal papillary cell carcinoma, stomach adenocarcinoma, and PDAC. Our study supports the significance of KYNU as a tumor marker of tumor immunosuppression and as a multi-cancer prognostic marker for poor overall survival.

**Abstract:**

The nuclear factor erythroid 2-related factor 2 (NRF2) pathway is frequently activated in various cancer types. Aberrant activation of NRF2 in cancer is attributed to gain-of-function mutations in the NRF2-encoding gene *NFE2L2* or a loss of function of its suppressor, Kelch-like ECH-associated protein 1 (*KEAP1*). NRF2 activation exerts pro-tumoral effects in part by altering cancer cell metabolism. Previously, we reported a novel mechanism of NRF2 tumoral immune suppression through the selective upregulation of the tryptophan-metabolizing enzyme kynureninase (KYNU) in lung adenocarcinoma. In the current study, we explored the relevance of NRF2-mediated KYNU upregulation across multiple cancer types. Specifically, using a gene expression dataset for 9801 tumors representing 32 cancer types from The Cancer Genome Atlas (TCGA), we demonstrated that elevated KYNU parallels increased gene-based signatures of NRF2-activation and that elevated tumoral KYNU mRNA expression is strongly associated with an immunosuppressive tumor microenvironment, marked by high expression of gene-based signatures of Tregs as well as the immune checkpoint blockade-related genes CD274 (PDL-1), PDCD1 (PD-1), and CTLA4, regardless of the cancer type. Cox proportional hazard models further revealed that increased tumoral KYNU gene expression was prognostic for poor overall survival in several cancer types, including thymoma, acute myeloid leukemia, low-grade glioma, kidney renal papillary cell carcinoma, stomach adenocarcinoma, and pancreatic ductal adenocarcinoma (PDAC). Using PDAC as a model system, we confirmed that siRNA-mediated knockdown of NRF2 reduced KYNU mRNA expression, whereas activation of *NFE2L2* (the coding gene for NRF2) through either small-molecule agonists or siRNA-mediated knockdown of *KEAP1* upregulated KYNU in PDAC cells. Metabolomic analyses of the conditioned medium from PDAC cell lines revealed elevated levels of KYNU-derived anthranilate, confirming that KYNU was enzymatically functional. Collectively, our study highlights the activation of the NRF2–KYNU axis as a multi-cancer phenomenon and supports the relevance of tumoral KYNU as a marker of tumor immunosuppression and as a prognostic marker for poor overall survival.

## 1. Introduction

Activation of the nuclear factor erythroid 2-related factor 2 (NRF2) pathway is a frequent finding in various cancer types. Aberrant activation of NRF2 in cancer is attributed to either gain-of-function mutations in the NRF2 encoding gene *NFE2L2* or loss of function of its suppressor, Kelch-like ECH-associated protein 1 (*KEAP1*). NRF2 activation exerts pro-tumoral effects in part by maintaining redox homeostasis and by altering cancer cell metabolism via regulating the expression and function of several key metabolic enzymes including those involved in glycolysis, pentose phosphate pathways, folate metabolism, tricarboxylic acid cycle (TCA), and glutaminolysis [1,2].

Recently, we reported an unrecognized role of NRF2 activation through loss-of-function *KEAP1* mutations in upregulating the tryptophan-metabolizing enzyme kynureninase (KYNU) in lung adenocarcinoma (LUAD) [3]. We showed that KYNU was enzymatically functional in LUAD cells and that elevated tumoral KYNU was profoundly associated with an immunosuppressive tumor microenvironment, characterized by increased regulatory T-cells and increased expression of immune checkpoint blockade targets including PD-L1. We further showed that increased tumoral KYNU expression was a prognostic predictor of poor overall survival in multiple independent LUAD transcriptomic datasets and tissue microarrays [3].

Given the frequency of NRF2 activation in multiple cancers, we sought to evaluate if KYNU upregulation with immune suppression was also seen across different cancer types. Leveraging gene expression data for 9801 tumors representing 32 cancer types from the Cancer Genome Atlas (TCGA), we demonstrated that elevated KYNU parallels increased gene-based signatures of NRF2 activation and that elevated tumoral KYNU mRNA expression is strongly associated with an immunosuppressive tumor microenvironment regardless of the cancer type. We further demonstrated that elevated tumoral KYNU gene expression is prognostic for poor overall survival in several cancer types, including pancreatic cancer (PDAC). Using PDAC as a model system, we showed that KYNU is enzymatically functional in PDAC cells and that KYNU is regulated by NRF2.

## 2. Materials and Methods

### 2.1. Cell Culture and Transfection

Cancer cell lines were maintained in RPMI medium plus 10% fetal bovine serum (FBS) unless otherwise stated. Small interfering RNA (siRNA) transfection experiments were performed using the following siRNAs: siControl (Silencer Select Negative Control #1, Life Technologies, Carlsbad, CA), siKEAP1 #1 and #2 (#00080908 and #00344034, Sigma Aldrich, St. Louis, MO, USA), and siNFE2L2 #1 and #2 (#00182393 and # 00341015, Sigma Aldrich).

### 2.2. Chemicals

CDDO-Me (2-Cyano-3,12-dioxo-oleana-1,9(11)-dien-28-oic acid methyl, or bardoxolone methyl) was purchased from Sigma Aldrich (Cat# SMB00376). Stock solutions were resuspended in dimethyl sulfoxide (DMSO).

### 2.3. RT-PCR Analysis

RNA was extracted using the RNeasy Extraction Kit (Qiagen, Hilden, Germany) according to manufacturer’s protocol. The TaqMan PCR assay was performed with a 7500 Fast Real-Time PCR System using the universal TaqMan PCR master mix (ThermoFisher, Waltham, MA, USA) and FAM^TM^-labeled probes for KYNU (Hs00187560_m1) and KEAP1 (Hs00202227_m1) and VIC^TM^-labeled probes for β2M (Hs_00187842_m1). PCR was carried out using a BioRad CFX Connect RT System. Values are reported as 2^−∆∆Ct^.

### 2.4. Proteomic Analyses

For proteomic analyses, each pancreatic cancer cell line (*n* = 11) was analyzed as a singular replicate using established operating procedures [3,4,5]. For these experiments, the cancer cell lines were grown for seven passages in RPMI-1640 supplemented with ^13^C-lysine and 10% dialyzed fetal bovine serum (FBS) according to the standard SILAC protocol [6]. The purpose of SILAC labeling was to discriminate cancer-derived proteins from FBS-derived proteins.

### 2.5. Metabolomic Analyses

The exometabolome experiments were performed on conditioned media from 11 PDAC cell lines collected at prespecified incubation times (baseline, 1, 2, 4, and 6 h) as described in our prior publication [7,8].

Metabolomics analysis for KP-related metabolites was conducted on a Waters Acquity™ 2D/UPLC system with parallel column regeneration configuration using *H*-class quaternary solvent manager and *I*-class binary solvent manager coupled to a Xevo G2-XS quadrupole time-of-flight (qTOF) mass spectrometer as previously described [3]. The mass spectrometry data were acquired on Xevo G2 XS qTOF in ‘sensitivity’ mode and in positive electrospray ionization (ESI) mode within the 50–1200 Da range. The LC–MS and LC–MSe data were processed using Progenesis QI (Nonlinear, Waters), and the values are reported as area units. Annotations for tryptophan, kynurenine (KYN), anthranilate, and 3-hydroxyanthranilate were determined by matching accurate mass and retention times using authentic standards and by matching experimental tandem mass spectrometry data against the NIST MSMS or HMDB v3 theoretical fragmentations.

To correct for injection order drift, each feature was normalized using data from repeat injections of quality control samples collected every 10 injections throughout the run sequence. The measurement data were smoothed by Locally Weighted Scatterplot Smoothing (LOESS) signal correction (QC-RLSC) as previously described [8]. The values are reported as normalized area units per hour per 100 ug of protein.

### 2.6. Gene Expression Datasets

Gene expression data, mutational information, and clinical data available for 9801 tumors representing 32 tumors types from The Cancer Genome Atlas (TCGA) network project were download from cbioportal (available online: http://www.cbioportal.org/, accessed date: 25 April 2022) [9]. Gene expression data and associated clinical information for the Badea Pancreas Study [10] were downloaded from the Oncomine database [11]. Gene expression data for 1037 cancer cell lines were downloaded from the Cancer Cell Line Encyclopedia (CCLE) (available online: https://sites.broadinstitute.org/ccle/, accessed date: 22 April 2022) [12]).

The consensus 11-gene NRF2-activation signature (SRXN11, NQO1, SLC7A11, PGD, OSGIN1, GCLM, CYP4F11, AKR1C3, AKR1B10, ABCC2, and NROB1) was based on reportings by Best et al. [13]. The gene-based signatures of immune cell infiltrates were based on studies by Bindea et al. [14].

### 2.7. Immunohistochemical Analyses

Representative immunohistochemical analyses of KYNU protein expression in pancreatic tumors were derived from The Human Protein Atlas (available online: https://www.proteinatlas.org/ accessed date: 30 June 2022). Staining was performed using a rabbit polyclonal anti-KYNU antibody (HPA031686 Atlas Antibodies, Bromma, Sweden).

### 2.8. Statistical Analyses

Statistical significance for two-class comparisons was determined using 2-sided Wilcoxon Rank Sum Tests unless otherwise specified. Spearman correlation analyses were used for the comparison of continuous variables. Cox proportional hazard models and construction of Kaplan–Meier survival curves were carried out in R statistical software. Cox proportional hazard models were used to evaluate the associations between KYNU mRNA expression levels and overall survival. Variables included into the multivariable Cox proportional hazard models were based on the backward stepwise method (likelihood ratio). To test for the proportionality of hazard assumption of a Cox regression, we used the method of Patricia and Grambsch [15].

Significance in survival distributions in Kaplan–Meier survival curves was determined by the Mantel-Cox Log-Rank *T*-test. For the survival analyses, we used the method described by Contal and O’Quigley [16] to derive an optimal change point for KYNU expression that yielded the largest difference between individuals in the two already defined groups (alive/dead) [8,17].

Figures were generated in either GraphPad Prism v6 or R statistical software.

## 3. Results

### 3.1. Association between Tumoral KYNU Expression and NRF2 Activation across Multiple Cancer Types

Using transcriptomic and genomic data available for 9801 tumors representing 32 different cancer types from The Cancer Genome Atlas (TCGA) PanCancer Atlas Studies, we assessed for associations between KYNU mRNA expression and presence of mutations in *NFE2L2*, the gene encoding for NRF2, or *KEAP1.* Of the 9801 analyzed tumors, 2.73% had *NFE2L2* mutations, 2.67% had *KEAP1* mutations, and 5.24% had either an *NFE2L2* or a *KEAP1* mutation. Differential analyses revealed KYNU mRNA levels to be statistically significantly (Wilcoxon rank sum test 2-sided *p* < 0.0001) elevated in tumors harboring *NFE2L2* and/or *KEAP1* mutations compared to tumors wild-type for *NFE2L2/KEAP1* (Figure 1A; Appendix A). Spearman correlation analyses between tumoral KYNU mRNA expression and an 11-gene NRF2 activation consensus signature [18] similarly revealed a positive correlation (overall Spearman *ρ*: 0.45 (95% CI: 0.44–0.47); 2-side *p* < 0.0001) (Figure 1B,C) [18].

The evaluation of transcriptomic data for 1037 cancer cell lines representing 22 different anatomical sites from the Cancer Cell Line Encyclopedia (CCLE) also showed KYNU mRNA levels to be positively correlated (Spearman *ρ*: 0.36 (95% CI: 0.31–0.42); 2-sided *p* < 0.0001) with the 11-gene NRF2 activation consensus signature (Figure 1D,E).

### 3.2. Association between NRF2 Activation and KYNU in Pancreatic Cancer Cells

PDAC was among the cancer types that exhibited a strong positive correlation between KYNU gene expression and the gene-based signature of NRF2 activation. Although mutations in *NRF2 (NFE2L2)* or *KEAP1* are uncommon in pancreatic cancer, NRF2 expression levels have been reported to be elevated in over 93% of PDAC [19].

To further explore the relationship between KYNU and NRF2 activation in PDAC, we initially assessed the mRNA levels of kynurenine pathway (KP) enzymes in PDAC tumors and adjacent control tissues using the Badea transcriptomic dataset [10]. Consistent with our prior findings in lung adenocarcinoma [3], the mRNA levels of KYNU, as well as several of the KP enzymes upstream of KYNU, were statistically significantly elevated (Wilcoxon rank sum test 2-sided *p* < 0.0001) in PDAC tumors compared to adjacent control tissues, whereas those of 3-hydroxyanthranilate 3,4-dioxygenase (HAAO), and quinolinate phosphoribosyltransferase (QPRT), enzymes involved in de novo NAD+ biosynthesis, were reduced (Figure 2A). Representative immunohistochemical analyses from the ProteinAtlas confirmed KYNU protein expression in PDAC tumors (Figure 2B). The evaluation of KP enzymes in the proteome of 11 pancreatic cancer cell lines previously profiled by mass spectrometry provided further evidence that KYNU is highly expressed in PDAC cells (Figure 2C).

To confirm that KYNU was enzymatically functional in PDAC cells, metabolomic analyses were performed on serially collected conditioned media (baseline, 30 min, 1 h, 2 h, 4 h, and 6 h) from PDAC cell lines. The analyses demonstrated a statistically significant positive correlation between the rate (area units/hour/100 ug of protein) of KYNU-derived anthranilate accumulation into the conditioned media of PDAC cell lines and cell lysate KYNU protein expression (Spearman *ρ*: 0.71 (95% CI: 0.17–0.92), 2-sided *p*: 0.02) (Figure 2D).

We next performed siRNA-mediated knockdown of NFE2L2 in KYNU-expressing SU.86.86 PDAC cells and confirmed that the loss of NRF2 reduced (2-sided *t*-test *p* < 0.05) KYNU mRNA levels (Figure 2E). Inversely, the activation of the NRF2 pathway through siRNA-mediated knockdown of KEAP1 or through treatment with the NRF2 agonist CDDO-Me in KYNU-low PANC1 PDAC cells resulted in marked increases in KYNU mRNA levels (Figure 2F).

### 3.3. Relationship between Tumoral KYNU Gene Expression, Tumor Immunophenotype, and Overall Survival across Different Cancer Types

Previously, we reported that elevated tumoral KYNU expression was associated with an immunosuppressive tumor microenvironment, characterized by increased signatures of regulatory T-cells (Tregs) as well as immune checkpoint blockade genes PD-1 and PD-L1 in LUAD and that elevated tumoral KYNU expression was prognostic for poor overall survival [3].

To explore the multi-cancer relevance of these findings, we assessed the association between tumoral KYNU mRNA levels with gene-based signatures of tumor immune cell infiltrates and immune checkpoint blockade-related genes across the 32 cancer types in TCGA transcriptomic datasets. The analyses indicated that cancer types with high KYNU gene expression (>median) generally had elevated gene-based signatures of tumor immune cell infiltrates, including Tregs, as well as immune checkpoint blockade-related genes CD274 (PD-L1), PDCD1 (PD-1), and CTLA4 (Figure 3A). These findings were replicated when restricting the analyses to individual cancer types, including PDAC (Figure 3B–D; Appendix A).

We further evaluated the association between tumoral KYNU mRNA expression and overall survival for the 32 cancer types in TCGA datasets. In addition to LUAD, univariable Cox proportional hazard models demonstrated elevated tumoral KYNU gene expression to be prognostic for poor overall survival in thymoma (THYM), acute myeloid leukemia (LAML), low-grade glioma (LGG), kidney renal papillary cell carcinoma (KIRP), stomach adenocarcinoma (STAD), and PDAC (Table A1 in the Appendix B). In multivariable Cox proportional hazard models, considering sex, age, and tumor stage or grade as co-variables, elevated tumoral KYNU mRNA expression yielded hazard ratios (HR) per log_10_ unit increase of 3.77 (95% CI: 1.64–8.70), 1.60 (95% CI: 1.00–2.50), 2.10 (95% CI: 1.41–3.20), and 2.00 (95% CI: 0.92–4.40) for PDAC, STAD, LGG, and KIRP, respectively (Figure 4). Multivariable Cox Proportional Hazard models for THYM and LAML were not performed due to limited clinical information. The assumptions of Cox proportional hazard were met in our models. Kaplan–Meier curves showed that elevated tumoral KYNU mRNA levels above an optimal cutoff point were associated with poor overall survival for THYM, LAML, KRIP, STAD, and PDAC (Figure 5).

## 4. Discussion

Increasing evidence supports a key role of NRF2 in reprogramming cancer cell metabolism, with several NRF2 target genes involved in the regulation of glycolysis, pentose phosphate pathway, fatty acid metabolism, glutamine metabolism, and glutathione metabolism [2,20,21]. Recently, we reported an unrecognized role of NRF2-mediated alterations in tryptophan metabolism through the selective upregulation of KYNU in LUAD that is associated with a tumor immunosuppression and poor prognosis [3]. In the current study, we provide evidence that NRF2-mediated KYNU upregulation extends beyond LUAD, as evidenced in the context of PDAC, and that the upregulation of the NRF2–KYNU axis is a frequent phenomenon across several cancer types. We further demonstrate that elevated tumoral KYNU mRNA expression is consistently associated with an immunosuppressive tumor microenvironment in multiple cancer types and that increased tumoral KYNU mRNA levels are prognostic for poor overall survival in THYM, LAML, LGG, KIRP, STAD, and PDAC.

The role of KP and altered tryptophan metabolites in promoting tumor immunosuppression has been well documented [22,23,24]. While early clinical trials of the small molecule inhibitor of IDO1, epacadostat, in combination with the anti-PD-1 molecule pembrolizumab, showed promise, no benefit was found in a phase III trial in metastatic melanoma patients compared to pembrolizumab alone [25,26,27,28]. Multiple studies have since reported considerable heterogeneity in KP enzyme expression, and there is increasing appreciation that other tryptophan catabolites can elicit immune suppressive effects [24,29,30]. For instance, KYNU-derived 3-hydroxyanthranilate (3-HAA) has been shown to inhibit NO production in macrophages, preventing macrophage-mediated tumor killing [31], induce HLA-G cell surface expression in dendritic cells, enhancing dendritic cells’ ability to tolerate antigens [32], induce apoptosis in both cytotoxic T-cells and T_h_1 populations of effector T cells [33,34], and enhance differentiation of Tregs [35,36].

Thus, our findings of elevated tumoral KYNU levels being frequently associated with tumor immune suppression and poor prognosis underscore several clinically relevant implications. For instance, the evaluation of circulating KYNU-derived metabolites may provide potential biomarker utility for predicting the response to immune checkpoint inhibition and/or for prognostication. To this end, elevated 3-HAA levels were found to be associated with poor progression-free survival in patients with non-small cell lung cancer (NSCLC) [37]. Notably, NSCLC patients that had elevated plasma 3-HAA levels and high tumoral PD-L1 expression had the highest objective response rates to immune checkpoint inhibitors [37]. In an independent study, elevated plasma levels of AA and, to a lesser extent, 3-HAA, were also associated with poor prognosis in NSCLC patients that subsequently received anti-PD-L1 treatment [38].

Targeting KYNU through small-molecule inhibition to combat tumor immune suppression represents an intriguing therapeutic strategy. To-date, a few small-molecule inhibitors of KYNU have been developed, including O-methoxybenzoylalanine, a KYN analogue [39], and benserazide hydrochloride [40]. However, these inhibitors have not been investigated in the context of cancer. Whether inhibitors of KYNU can reverse tumor immunosuppression remains an active area of exploration. Yet, targeting KYNU also garners some considerations. AA is derived from the catabolism of kynurenine (KYN) by KYNU, and, unlike 3-HAA, AA is thought to be immunologically inert. In immune-competent mouse models of melanoma, breast cancer, and colon cancer, PEGylated kynureninase (PEG-KYNase) administration was associated with marked reductions in serum KYN levels and increases in AA levels that were met with concomitant increases in tumor-infiltrating CD8+ T lymphocytes and tumor reduction. Anti-cancer efficacy was further enhanced when combining PEG-KYNase with immune checkpoint inhibitors [6]. Another study showed that the overexpression of KYNU in Chimeric Antigen Receptor (CAR)-T cells depleted the immunosuppressive KYN levels in the tumor microenvironment, resulting in enhanced CAR-T-cell anti-cancer efficacy [41].

Collectively, these studies support that the catabolism of KYN to AA by KYNU promotes tumor immune infiltration, which is consistent with our observations that elevated levels of tumoral KYNU are associated with higher gene-based signatures of tumor immune cell infiltrates. However, elevated KYNU in the presence of NRF2 activation may tilt the balance from an anti-tumoral immune response towards T-cell exhaustion and differentiation into Tregs that is tumor promotive. In this regard, prior studies demonstrated that PD-L1+ *Keap1^f/f^/Pten^f/f^* (K1P) NSCLC tumor-bearing mice have elevated tumoral PD-L1 expression and increased presence of PD-1+ CD8+ T-lymphocytes compared to uninfected K1P or *Kras*^G12D^/*p53*^f/f^ tumor-bearing mice [18]. Moreover, K1P-tumor-bearing mice were responsive to anti-PD-L1 and anti-CTLA-4 treatment [18].

Radiation-induced NRF2 activation has also been shown to transcriptionally induce PD-L1 expression in melanoma [42]. Additionally, the presence of inflammatory stimuli may also alter the KYNU substrate affinity from KYN to 3-hydroxykynurenine (3HK), resulting in increased 3-HAA synthesis. A prior study reported that KYNU was highly upregulated in HER2-enriched and triple-negative breast cancer subtypes and that elevated KYNU levels were associated with increased levels of AA and 3-HAA [43]. Under basal conditions, breast cancer cells preferentially had high levels of AA synthesis. Stimulation with IFN-γ resulted in a pronounced 286-fold increase in 3-HAA synthesis [43]. Further investigations are needed to define the underlying mechanism(s) by which KYNU alone and in the presence of NRF2 activation modulates the tumor immune response and to establish the therapeutic potential of targeting tumor KYNU.

## 5. Conclusions

In conclusion, our findings demonstrate the upregulation of the NRF2–KYNU axis through multiple cancer types and highlight the relevance of tumoral KYNU as a marker of tumor immunosuppression and as a prognostic marker for poorer overall survival.

## Figures and Tables

**Figure 1 cancers-15-00834-f001:**
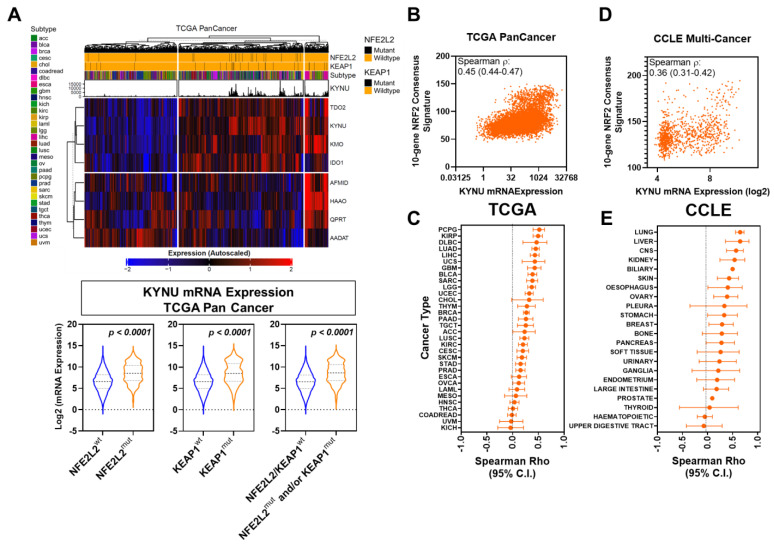
Association between KYNU gene expression and NRF2 status across different cancer types. (**A**) Unsupervised heatmap showing gene expression of kynurenine pathway (KP) enzymes and their association with *NFE2L2* and *KEAP1* mutational status across 9801 tumors representing 32 cancer types in the TCGA transcriptomic datasets. Violin plots illustrate the mRNA levels of KYNU among tumors stratified based on mutations in *NRF2 (NFE2L2)* and/or *KEAP1*. Statistical significance was determined by the 2-sided Wilcoxon rank sum test. (**B**) Scatter plot depicting the association between KYNU mRNA levels and a 10-gene NRF2 signature [3] across the 9803 tumors from TCGA. (**C**) Dot plot showing Spearman rho correlation coefficients (95% confidence interval) between tumoral KYNU mRNA levels and the 11-gene NRF2 signature across the 32 cancer types in the TCGA PanCancer Atlas dataset. (**D**) Scatter plot illustrating the relationship between KYNU mRNA levels and the 11-gene NRF2 signature [3] across the 1037 cancer cell lines from CCLE. (**E**) Dot plot depicting Spearman rho correlation coefficients (95% confidence interval) between KYNU mRNA levels and the 11-gene NRF2 signature across cancer cell lines stratified into 22 anatomical sites in CCLE.

**Figure 2 cancers-15-00834-f002:**
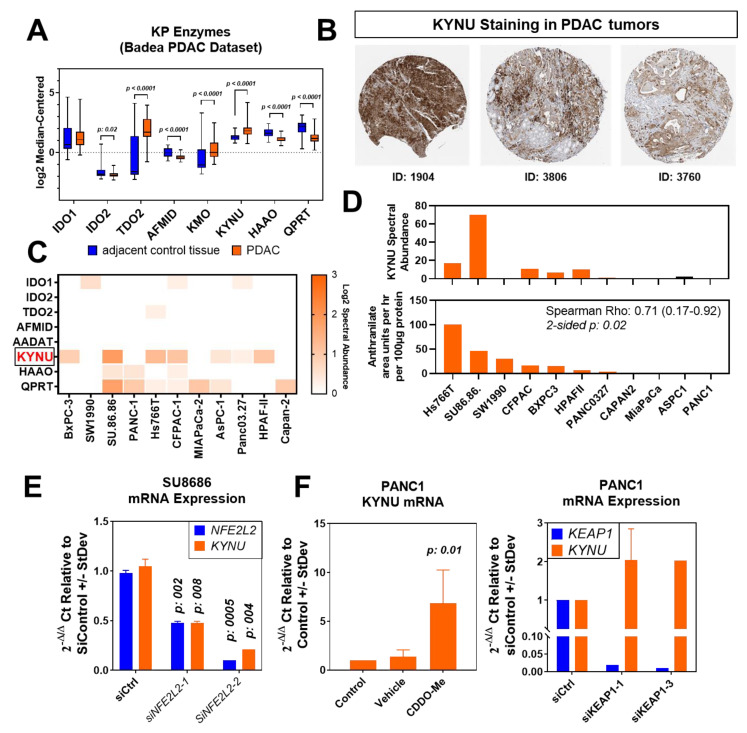
NRF2 regulates KYNU expression in pancreatic cancer cells. (**A**) mRNA expression of KP enzymes in pancreatic tumors and adjacent control tissues. The gene expression data were derived from the Badea study [10]. Statistical significance was determined by the 2-sided Wilcoxon matched-pairs signed rank test. (**B**) Representative immunohistochemical staining of KYNU in pancreatic tumors. Data were derived from the ProteinAtlas. (**C**) Protein abundance (spectral abundance) of KYNU in whole cell lysates of 11 pancreatic cancer cell lines analyzed by mass spectrometry. (**D**) Bar plots showing the association between whole-cell-lysate KYNU protein expression and rates (area units per hour per 500 µg of protein) of KYNU-derived anthranilate accumulation in the conditioned medium of the respective pancreatic cancer cell lines. (**E**) Relative mRNA expression of KYNU following siRNA-mediated knockdown of NFE2L2 in SU.86.86 pancreatic cancer cells. (**F**) Relative mRNA expression of KYNU following treatment of KYNU-low PANC1 pancreatic cancer cells with the NRF2 agonist CDDO-Me (left panel) or following siRNA-mediated knockdown of *KEAP1* (right panel).

**Figure 3 cancers-15-00834-f003:**
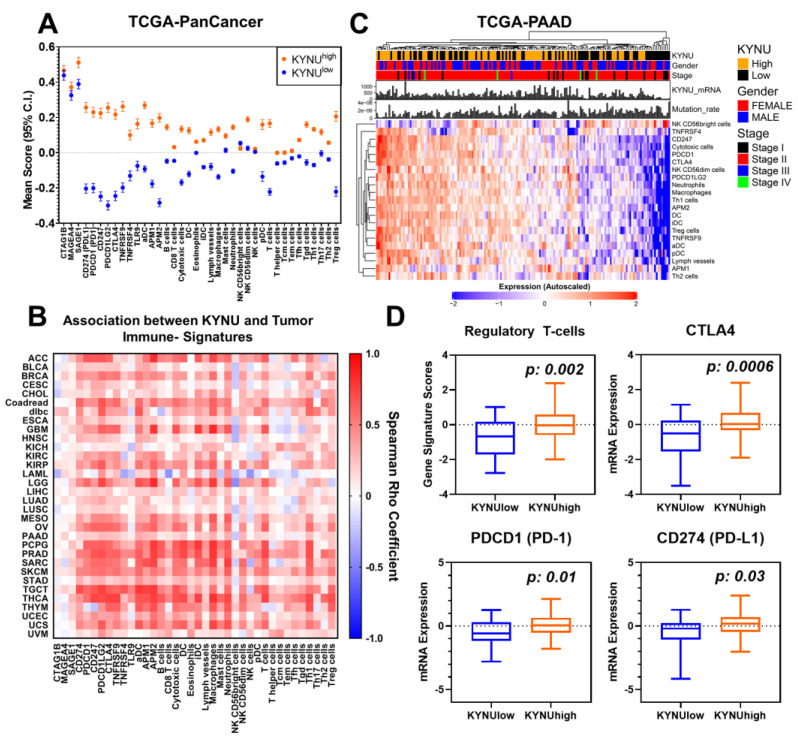
Association between tumoral KYNU mRNA expression and tumor immunophenotype. (**A**) Gene-based signature scores for immune cell infiltrates and immune checkpoint blockade-related genes across 9801 tumors from TCGA stratified into high (>median) or low (≤median) KYNU mRNA levels. For these analyses, cancer type was not considered. The immune signature scores were based on Bindea et al. [14] (**B**) Spearman correlation heatmap illustrating the association between tumoral KYNU mRNA levels and gene-based signatures of immune subtypes and immune checkpoint blockade genes across the 32 cancer types. (**C**) Heatmap depicting the relationship between KYNU mRNA expression and gene-based signatures of immune subtypes in the TCGA-PDAC dataset. Immune subtypes that were significantly differential (Student’s *t*-test 2-sided *p* < 0.05) between KYNU-high (>median) and KYNU-low (≤median) tumors are shown. (**D**) Box and whisker plots showing gene signature scores for Tregs and mRNA levels of CTLA4, PDCD1 (PD-1), and CD274 (PD-L1) in PDAC tumors with KYNU mRNA levels in the bottom 25th percentile and top 25th percentile.

**Figure 4 cancers-15-00834-f004:**
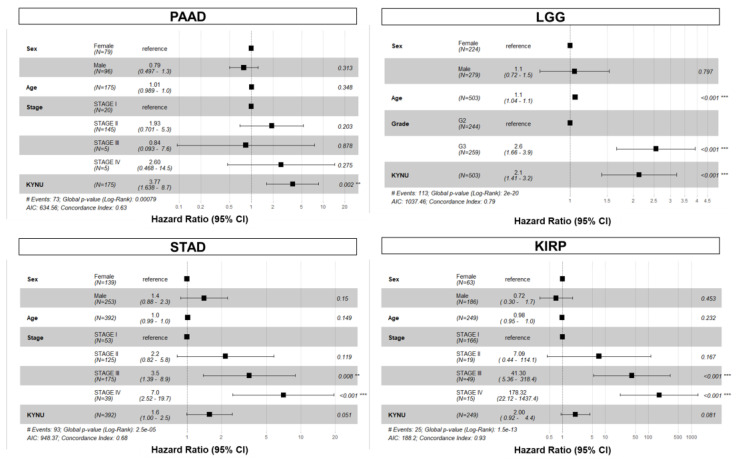
Forest plots depicting hazard ratios (HR) and 95% confidence intervals for pertinent clinical variables and tumoral KYNU mRNA levels (por log_10_ unit increase) for overall survival in low-grade glioma (LGG), kidney renal papillary cell carcinoma (KIRP), stomach adenocarcinoma (STAD), and pancreatic adenocarcinoma (PDAC) TCGA datasets.

**Figure 5 cancers-15-00834-f005:**
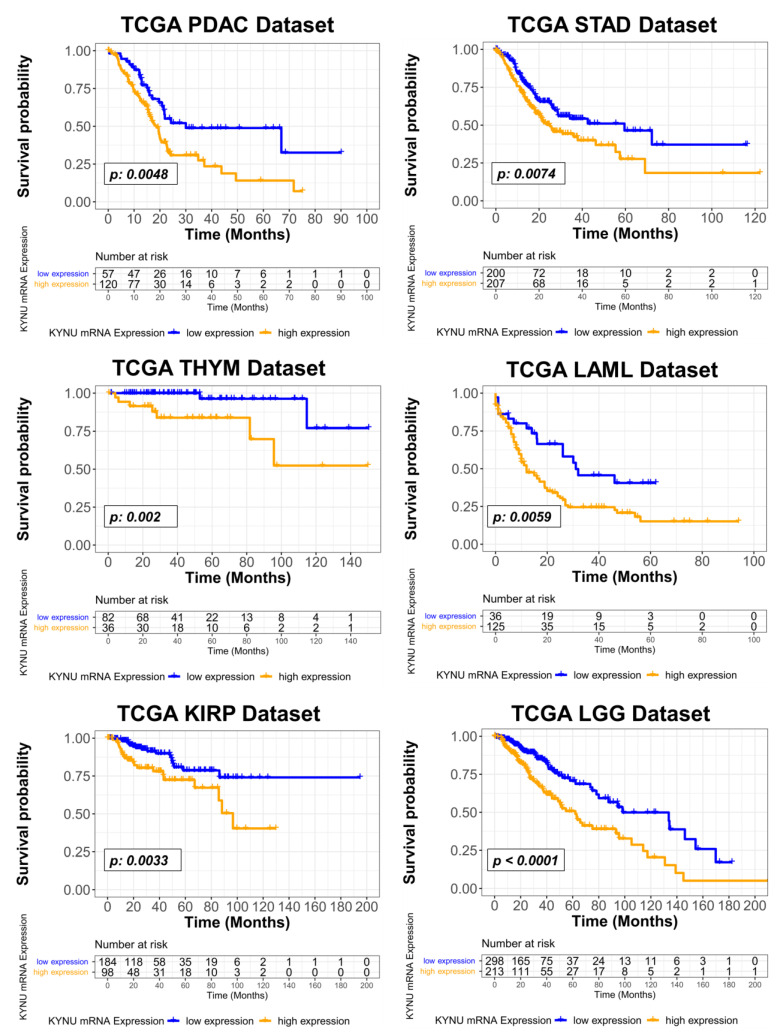
Kaplan–Meier curves for overall survival in PDAC, stomach adenocarcinoma (STAD), kidney renal papillary cell carcinoma (KIRP), thymoma (THYM), acute myeloid leukemia (LAML), and low-grade glioma (LGG) TCGA datasets based on KYNU mRNA levels at an optimal cutoff value. The optimal KYNU cutoff was derived using the method described by Contal and O’Quigley [16]. Statistical significance was determined by two-sided log-rank Mantel–Cox test.

## Data Availability

Relevant data supporting the findings of this study are available within the Article and Appendices or are available from the author upon reasonable request.

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
