# Peer review of "Kynureninase Upregulation Is a Prominent Feature of NFR2-Activated Cancers and Is Associated with Tumor Immunosuppression and Poor Prognosis"

_cancers, 2023, doi:10.3390/cancers15030834_

Round 1

Reviewer 1 Report

The manuscript by Leon-Letelier et al focuses on the link between NRK2, kynureninase and immunosuppression in pancancer data sets. The report is a follow up to previous studies by the group that detected the association in lung adenocarcinoma and here finds a similar relationship in pancreatic ductal, thymoma, AML and several other tumors. siRNA-mediated manipulation of expression in PDAC cell lines supports the relationship.

The NRK2-kynureninase link has been reported previously as has the kynureninase- immune suppression link. However, the pancancer study reveals associations in tumor types not previously categorized in this way. Appropriate statistical analyses support the conclusions. Hence, the data presented is informative and the manuscript is worthy of publication.

Minor syntax and spelling errors are evident and should be corrected.

Author Response

Reviewer #1

The manuscript by Leon-Letelier et al focuses on the link between NRF2, kynureninase and immunosuppression in pancancer data sets. The report is a follow up to previous studies by the group that detected the association in lung adenocarcinoma and here finds a similar relationship in pancreatic ductal, thymoma, AML and several other tumors. siRNA-mediated manipulation of expression in PDAC cell lines supports the relationship.

The NRK2-kynureninase link has been reported previously as has the kynureninase- immune suppression link. However, the pancancer study reveals associations in tumor types not previously categorized in this way. Appropriate statistical analyses support the conclusions. Hence, the data presented is informative and the manuscript is worthy of publication.

Minor syntax and spelling errors are evident and should be corrected.

Response: We thank the Reviewer for their support of our findings. We have carefully gone through the manuscript to correct any syntax or spelling errors.

Reviewer 2 Report

This manuscript shows KINU as a potential biomarker for immunodepression in sevral types of cancers. This is a very interesting line of research and the workshowed here have a high significance of content, as the KP have recently become one of the most studied pathways in cancer. However, I do want to suugest a couple of modifications:

1. In figure 2B in would be very useful if the authors could highlight KYNU. 

2. In the discussion, it is mentioned that KYNU could serve as a potential biomarker and also could be a potential target for future treatments. Authors could explain further this idea, since it would be very helpful to target immunosupression. 

3. Authors need to correct the space between foot notes and the main text.  Sometimes is very difficult to know where the main text ends and the foot note starts (line 291)

Author Response

This manuscript shows KINU as a potential biomarker for immunodepression in sevral types of cancers. This is a very interesting line of research and the workshowed here have a high significance of content, as the KP have recently become one of the most studied pathways in cancer. However, I do want to suggest a couple of modifications:

  1. In figure 2B in would be very useful if the authors could highlight KYNU. 

Response: We have now modified Figure 2B to highlight KYNU.

  1. In the discussion, it is mentioned that KYNU could serve as a potential biomarker and also could be a potential target for future treatments. Authors could explain further this idea, since it would be very helpful to target immunosupression. 

Response: To the best of our knowledge, developed small molecule inhibitors of KYNU are O-Methoxybenzoylalanine, a KYN analogue, and benserazide hydrochloride. These inhibitors have been largely explored in the context of neurotoxicity and neurochemical and behavior effects, respectively. However, these inhibitors have not yet been investigated in the context of cancer. Whether targeting of KYNU via o-methoxybenzoylalanine or benserazide hydrochloride can reverse tumor immunosuppression remains an active area of exploration.

We have now commented upon this in the discussion section of the revised manuscript.

  1. Authors need to correct the space between foot notes and the main text.  Sometimes is very difficult to know where the main text ends and the foot note starts (line 291)

Response: We correct the space between foot notes and the main text.